

# Efficient adaptive Bayesian estimation of a slowly fluctuating Overhauser field gradient

Jacob Benestad[1], Jan A. Krzywda[2], Evert van Nieuwenburg[2] and Jeroen Danon[1*]

**1** Center for Quantum Spintronics, Department of Physics,
Norwegian University of Science and Technology,
NO-7491 Trondheim, Norway
**2** Lorentz Institute and Leiden Institute of Advanced Computer Science,
Leiden University, P.O. Box 9506, 2300 RA Leiden, The Netherlands

* jeroen.danon@ntnu.no

## Abstract

Slow fluctuations of Overhauser fields are an important source for decoherence in spin qubits hosted in III-V semiconductor quantum dots. Focusing on the effect of the field gradient on double-dot singlet–triplet qubits, we present two adaptive Bayesian schemes to estimate the magnitude of the gradient by a series of free induction decay experiments. We concentrate on reducing the computational overhead, with a real-time implementation of the schemes in mind. We show how it is possible to achieve a significant improvement of estimation accuracy compared to more traditional estimation methods. We include an analysis of the effects of dephasing and the drift of the gradient itself.

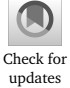

# 1   Introduction

Spin-based semiconductor devices offer several very useful properties for hosting qubits, including their small size, long relaxation times, fast gate-operation times, and a good potential for scalability based on their similarity to conventional electronic devices [1–3]. Initially, research focused on type III-V semiconductors, and particularly GaAs, because there is no valley degeneracy in the conduction band and heterostructure engineering was further developed than for other materials. However, soon it was realized that for III-V semiconductors hyperfine coupling of the localized spins to the nuclear spin baths cannot be avoided, and the resulting randomly fluctuating Overhauser fields limit such devices to very short spin dephasing times $T_2^* \sim 20$ ns [4–6]. Indeed, it was the eventual development of devices based on materials that can be isotopically purified to be almost nuclear-spin-free, such as Si and Ge [7–13], that propelled a recent leap in performance for spin qubits, providing high fidelity and long coherence times which allowed for 4- and 6-qubit quantum logic with spin qubits [14, 15].

The harmful Overhauser field fluctuations are, however, very slow (typically on the scale of seconds), and an alternative approach could thus be to monitor these fluctuations in real time and adjust qubit control accordingly: accurate knowledge about the Overhauser fields can be used to significantly extend the qubit coherence time [16–18]. Furthermore, while universal control of spin qubits in materials with weak spin–orbit coupling has typically relied on the use of micromagnets [15, 19–21] or microwave striplines [5, 22, 23], the Overhauser fields can also be used as control axes, as long as they are known within sufficient uncertainty [24, 25].

The development of fast and reliable protocols for real-time estimation of Overhauser fields could thus lift some of the main limitations of spin qubits realized in type III-V semiconductor devices, but also allow the use of Si and Ge devices without the costly process of isotopic purification. Besides, such protocols can also be used to estimate other slowly fluctuating Hamiltonian parameters, such as the low-frequency components of charge noise [26], and thus eliminate their contribution to qubit decoherence. Although thus relevant in a much broader sense, we will focus here on Hamiltonian parameter estimation in the context of fluctuating Overhauser fields. More specifically, we will consider double-quantum-dot-based singlet–triplet qubits, where the Overhauser field gradient over the two dots $\Delta B_z$ is the most important fluctuating parameter to be estimated.

One powerful tool for quantum sensing and estimation is provided by Bayesian statistics [27], which can be used to optimize estimation procedures on-the-fly. Bayesian estimation schemes have already been used for estimating the Overhauser fields in GaAs-based spin qubits [17,18,25], although so far only in a non-adaptive way where the whole estimation procedure, based on a series of single-shot free induction decay experiments, is predetermined. Inspired by the availability of field-programmable gate arrays (FPGAs) which perform real-time data processing and control feedback [28–32], we investigate the feasibility of implementing fast and efficient adaptive Bayesian estimation of $\Delta B_z$ in a singlet–triplet spin qubit, keeping the state-of-the-art experimental equipment in mind as a boundary condition, both in terms of the limits on calculation complexity and information storage capacity.

Ideally, an adaptive Bayesian estimation scheme uses global optimization, in the sense of always considering all possible future measurements when deciding for the next set of parameters. Global optimization strategies are, however, hard to implement in an efficient way, and one thus usually reverts to a so-called greedy strategy, where only the optimization of the next single-shot experiment is considered. Although thus suboptimal, such greedy strategies have been shown to yield an exponential scaling of the estimation error as a function of the number of single-shot experiments [33, 34]. Exact implementation of the optimal greedy adaptive scheme is, however, still computationally too intensive for real-time feedback in most instances, and instead there are typically two options: (i) approximate the distribu-

tion of possible estimates such that simple parametric solutions are possible [34, 35], or (ii) use Monte-Carlo sampling and approximate the optimal experiment to perform using some heuristic [36–41]. Otherwise, simple analytical solutions are only attainable for specific problems [42, 43] or when the space of experimental design is sufficiently constrained [44].

In this paper, we focus on the first option, where $\Delta B_z$ is estimated based on the approximation that its probability distribution remains Gaussian throughout the whole procedure [34], the advantage being that this only requires working with two parameters (the mean and variance of the distribution). The main challenge with this approach is to fit the posterior distribution after each measurement to a Gaussian in a computationally efficient way. We propose two methods that are simple enough to implement on a state-of-the-art FPGA and improve on existing schemes in that they allow for any Gaussian prior, including priors with small mean compared to their standard deviation. We first present a scheme where the fitting is based on the method of moments, for which we derive an efficient implementation that only relies on few straightforward calculations, paying particular attention to the problem of how to handle distributions of $\Delta B_z$ that are centered around zero [34]. Secondly, we explore the possibility of using a neural network (NN) to replace the parametric update equations for the Gaussian mean and variance, as it has been shown that NNs can be applied for tasks like finding the optimal design of experiments [45], updating the parameter distribution [46], and predicting the Hamiltonian at future times [47]. Finally, we show how the Gaussian approximation can also allow straightforwardly to account for the Overhauser-field dynamics in between estimations, thus adding a component of prediction to the schemes.

The structure of the rest of this paper is as follows. In Section 2 we introduce the basic physics of singlet–triplet qubits, focusing on the role of the Overhauser gradient, and Section 3 introduces the rationale behind Bayesian estimation and presents the specifics of the two schemes we propose. Then, in Section 4.1, we present numerical simulations of the two estimation schemes, benchmarking them against a more standard non-adaptive approach, both with and without a finite phenomenological dephasing time $T$. In Section 4.2 we analyze how a slow drift of the parameter to be estimated limits the number of useful measurements that can be performed, and how this can be related to the dephasing time $T$. Finally, in Section 4.3 we consider how the evolution of a Gaussian distribution in the Fokker–Planck formalism makes our schemes predictive, allowing for fewer measurements in future estimations.

## 2 System dynamics

Below we will discuss Bayesian estimation protocols for both static and slowly fluctuating Hamiltonian parameters, in relatively general terms. The specific system we will have in mind throughout is a two-electron singlet–triplet spin qubit hosted in a double quantum dot defined in a III–V-based semiconductor heterostructure. In this Section, we will highlight the relevant part of the physics of this system.

Singlet–triplet qubits are usually hosted in double quantum dots tuned to a $(1,1)$ charge configuration. In that regime, the gate-tunable exchange interaction $J(\epsilon)$ controls the qubit splitting, and a randomly fluctuating Overhauser field gradient drives rotations around the $x$-axis on the Bloch sphere. The two-level qubit Hamiltonian can be approximated as

$$H = \frac{\hbar \omega(t)}{2}\sigma_x + \frac{J(\epsilon)}{2}\sigma_z\,, \tag{1}$$

where $\sigma_{x,z}$ are Pauli matrices in the qubit basis $\{|0\rangle,|1\rangle\}$ and $\hbar\omega(t) = g\mu_B[B_z^{(1)}(t) - B_z^{(2)}(t)]$ in terms of the fluctuating Overhauser fields $\mathbf{B}^{(1,2)}(t)$ on the two dots, with $g$ the effective electronic $g$-factor and $\mu_B$ the Bohr magneton (see Fig 1). In principle, the estimation scheme

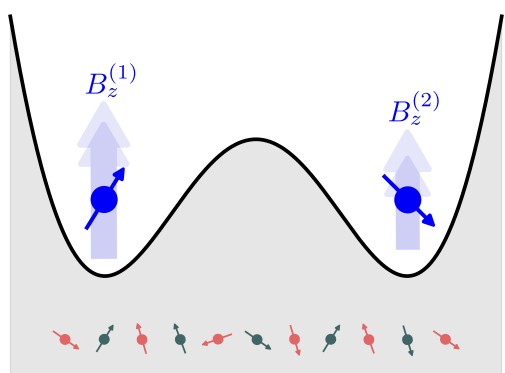

Figure 1: Due to the randomly fluctuating nuclear spins of the host material, the Overhauser fields acting on the two electron spins are unequal and slowly fluctuating. We model the dynamics of these fields as a drift–diffusion process, driven by rapid randomly occurring nuclear spin-flip processes.

presented below can be used to find the qubit frequency $\Omega(\epsilon, t) = \sqrt{\omega(t)^2 + J(\epsilon)^2/\hbar^2}$ for any detuning $\epsilon$. However, for simplicity we will concentrate on the case where all free qubit evolution takes place deep in the $(1,1)$ region, where $J(\epsilon) \approx 0$ and we thus estimate $\Omega(t) \approx |\omega(t)|$. The tunable exchange splitting is only made non-zero for initialization and readout purposes in this case.

We assume that the two Overhauser fields and thus the field gradient $\omega(t)$ follow an OU process [48]. The slow fluctuations of the effective fields arise from averaging the nuclear spin polarization dynamics of the typically $10^5$–$10^6$ nuclei that surround the electrons localized in the quantum dots, as illustrated in Fig 1. The dynamics of the OU process are compatible with the microscopic picture of a classical birth–death process, where random nuclear spin flips occur with a fixed rate (picturing, for simplicity, the nuclei to be Ising spins). This results in a net diffusion of $\omega(t)$ with an entropic drift towards zero. Working with a probability distribution for $\omega$, the dynamics of such a drift–diffusion process can be approximated by a Fokker–Planck equation. For an initial probability distribution that is Gaussian, with average $\mu(0)$ and variance $\sigma(0)^2$, solving the Fokker–Planck equation yields the time-dependent distribution

$$p(\omega, t) = \frac{1}{\sqrt{2\pi\sigma(t)^2}} \exp\left\{ -\frac{[\omega - \mu(t)]^2}{2\sigma(t)^2} \right\}, \tag{2}$$

which is a Gaussian distribution with a time-dependent mean $\mu(t) = \mu(0)e^{-\Gamma t}$ and variance $\sigma(t)^2 = \sigma_K^2 + [\sigma(0)^2 - \sigma_K^2]e^{-2\Gamma t}$. The parameters $\sigma_K$ and $\Gamma$ follow from the drift and diffusion constants and can be interpreted as the steady-state r.m.s. value of the Overhauser field gradient ($\sigma_K \sim 50$ MHz typically) and the slow relaxation rate of nuclear spin polarization ($\Gamma \sim 0.2$ Hz typically). We note that the inverse, $T_c = \Gamma^{-1} \sim 5$ s, sets the correlation time scale of the fluctuations of $\omega$, which defines the scale of the time window within which a single estimation of $\omega(t)$ is useful since the value would have drifted enough so that all potential information gain is lost.

## 3 Adaptive Bayesian estimation

We assume that the tool we have available for probing the Overhauser gradient is so-called free induction decay (FID) experiments: (i) the qubit is initialized in the basis state $|0\rangle$ (at the north

pole of the Bloch sphere), then (ii) it is left to precess freely around the field $\omega(t)$ that points along the $x$-axis on the Bloch sphere, for a time $\tau$, after which (iii) the qubit state is projectively measured in the qubit basis $\{|0\rangle, |1\rangle\}$, i.e., along the Bloch sphere's $z$-axis. The outcome (0 or 1) of every such experiment contains information about the precession frequency $\omega(t)$, and the goal is to find an optimal set of FID experiments (a set of "waiting times" $\{\tau_n\}$, with $n$ the index tracking the number of FID experiments) from which accurate knowledge about $\omega(t)$ can be distilled as efficiently as possible.

For this purpose, we use a Bayesian estimation approach where our knowledge about $\omega(t)$ is encoded in a probability distribution $p(\omega)$, which gets updated after each FID experiment according to Bayes' rule,

$$p_n(\omega|d_n, \tau_n, T) \propto p_{n-1}(\omega) p(d_n|\omega, \tau_n, T). \tag{3}$$

Here, $p_{n-1}(\omega)$ is the prior (old) distribution of $\omega$ and $p_n(\omega|d_n, \tau_n, T)$ the posterior (new) distribution, which is a compromise between our prior knowledge of $\omega$ and the new data point $d_n$ obtained, taking into account $\tau_n$ as well as the model parameter $T$ (see below) [49]. In our case, the data points are the binary measurement outcomes $d_n \in \{0, 1\}$ of the FID experiment, labeling the two qubit states. The so-called likelihood function—the probability to measure $d$ for given $\omega$, $\tau$, and $T$—is given by the Born rule

$$p(d|\omega, \tau, T) = \frac{1}{2}\left[1 + (-1)^d e^{-\tau^2/T^2} \cos(\omega\tau)\right], \tag{4}$$

where we included a phenomenological "dephasing time" $T$ that limits the coherence of a single-shot measurement and subsequently defines the longest useful waiting time $\tau_n$ for each FID experiment. Indeed, for $\tau \gtrsim T$ the likelihood function quickly reduces to $p = \frac{1}{2}$ for both $d$, independent of the other parameters, meaning that no information can be gained from the experiment. In many cases, the appropriate value to insert for $T$ can also be estimated from experiments in a Bayesian fashion [45, 50].

Before the first FID experiment is performed, i.e., when we have no information about $\omega$ at all, we assume a Gaussian probability distribution,

$$p_0(\omega) = \frac{1}{\sqrt{2\pi\sigma_K^2}} \exp\left\{-\frac{\omega^2}{2\sigma_K^2}\right\}, \tag{5}$$

corresponding to the steady-state limit of Eq. (2). We then see from Eqs. (3,4) that, independently of the choice of $\{\tau_n\}$ and $T$ and of the measurement outcomes $d_n$, every subsequent iteration of the probability distribution will be symmetric in $\omega$, i.e., $p_n(\omega) = p_n(-\omega)$. This is a consequence of the projective nature of the measurements; the direction of precession around the $x$-axis on the Bloch sphere is impossible to distinguish with FID experiments such as those performed here. In this sense, the best we can achieve is an accurate estimate of $|\omega|$. However, we note that if it is possible to control the cosine phase in Eq. (4) it would be possible to distinguish the sign of $\omega$.

We now turn to the question how to choose the best set of waiting times $\{\tau_n\}$. An important feature of Bayesian estimation is that it allows for "on-the-fly" optimization of free parameters: For each new experiment an optimal time $\tau_n$ can be computed, based on the current distribution function, in order to gain the maximum amount of information about $\omega$ [51]. There are several ways to quantify such information gain, the change in information entropy being the canonical choice [52]; yet in order to make calculations feasible to implement on an FPGA in real time, it would be easier to consider a simple quantity such as the variance of the distribution, using its degree of minimization during the estimation procedure as a measure for success.

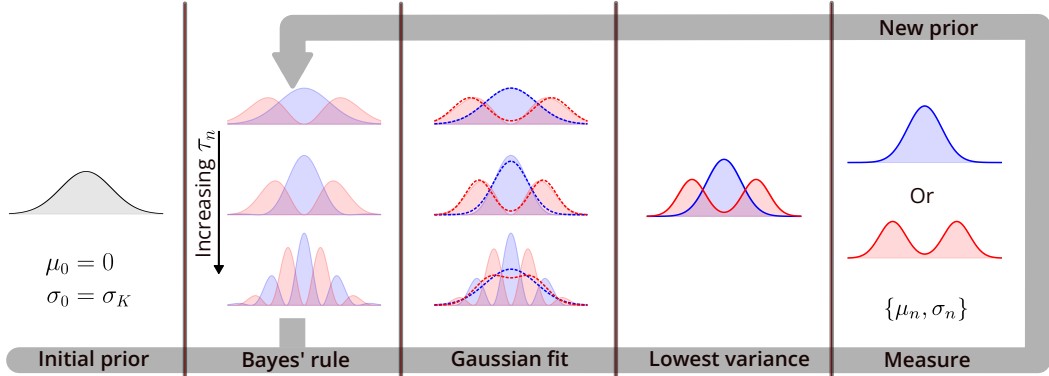

Figure 2: The Bayesian estimation cycle adapted here: To a bimodal Gaussian prior we apply Bayes' rule for many different $\tau$ and both potential outcomes $d \in \{0,1\}$. All resulting posteriors are fitted to a bimodal Gaussian again and the $\tau$ yielding the lowest posterior expectation value for $\sigma$ is selected. Finally, the experiment is performed, which determines the new prior.

However, since our distribution function is always symmetric in $\omega$, a straightforwardly calculated variance of $p(\omega)$ will in general not be a good measure for the uncertainty in $|\omega|$. We address this problem with the next simplification we make: Throughout the whole estimation procedure, we will always approximate posterior distributions $p(\omega|d, \tau, T)$ with a bimodal Gaussian,

$$q(\omega, \mu, \sigma) = \frac{1}{\sqrt{2\pi\sigma^2}} \exp\left\{-\frac{\omega^2 + \mu^2}{2\sigma^2}\right\} \cosh\left(\frac{\omega\mu}{\sigma^2}\right), \tag{6}$$

where $\pm\mu$ and $\sigma^2$ parameterize the mean and variance of the two Gaussian peaks, respectively, thus taking care of the indistinguishability of the sign of $\omega$ [34]. In the context of real-time Overhauser gradient estimation, this approximation has several important advantages: (i) analytic expressions of the $\tau_n$-dependent posterior distributions are straightforward to evaluate, possibly allowing for efficient optimization of $\tau_n$ on an FPGA, (ii) the distribution function is fully characterized by only two parameters, $\mu$ and $\sigma$, throughout the whole estimation procedure, which can save significant memory as compared to storing the actual distribution function $p_n(\omega)$, and (iii) incorporating the (slow) dynamics of the Overhauser gradient is straightforward by implementing the time-dependence of $\mu$ and $\sigma$ as mentioned below Eq. (2).

Returning to the question how to choose the $\{\tau_n\}$, we see that a natural choice is to pick the waiting times in a way that minimizes $\sigma$ in the Gaussian approximation of the distribution function. Ideally, one follows an adaptive strategy that aims at every iteration for a path that results in a globally optimal solution, i.e., a minimal expected $\sigma$ for the final distribution after the last iteration. However, since this is difficult in practice [51], one must typically settle for a "greedy" approach where the optimization is only considering the next experiment, which is what we will do here.

We thus consider the estimation procedure sketched in Fig 2, which includes the following steps:

1. The prior $p_{n-1}(\omega)$ is forced to always be a bimodal Gaussian distribution.

2. Use Eq. (3) to calculate $p_n(\omega|d_n, \tau, T)$ for all relevant $\tau$ and for both potential outcomes $d_n \in \{0, 1\}$ (blue and red curves).

3. Fit both calculated posteriors at each candidate $\tau$ to the bimodal Gaussian form (dashed curves).

4. Select the $\tau$ whose fit yields the smallest expected $\sigma^2$, considering a weighted average over the two possible measurement outcomes.

5. Perform experiment $n$ with the selected $\tau_n$; the outcome determines the prior for the next iteration.

The main remaining question is how to fit the posteriors calculated from Bayes' rule to a bimodal Gaussian distribution in a computationally efficient way. Below we present two different fitting schemes we investigated.

## 3.1 Method of moments fit

Fitting a more complicated distribution to a single Gaussian peak is a common approximation in statistics, and computationally simple fits can be made using for instance Laplace's approximation [53] or the method of moments (MM). In our case, as explained above, we will always have a symmetric distribution function that we want to fit to a *bimodal* Gaussian, which in general requires some caution in designing the fitting procedure.

In this section, we will explore the use of the MM to fit our posterior distributions to Eq. (6), as the oscillating nature makes Laplace's approximation unsuited in this case since it relies on the curvature of the distribution mode. The simplest application of the MM amounts to finding the bimodal Gaussian distribution whose two lowest moments are equal to the corresponding moments of the distribution one wants to fit. The fact that our distributions are always symmetric causes all odd moments to vanish, which means that the moments we focus on are the second and fourth.

The raw moments $\langle\omega^2\rangle_n$ and $\langle\omega^4\rangle_n$ of the posterior distribution for the two possible measurement outcomes $d_n \in \{0, 1\}$ follow from calculating

$$\langle\omega^r\rangle_n = \int_{-\infty}^{\infty} d\omega\, \omega^r p_n(\omega|d_n, \tau, T),\tag{7}$$

where $p_n(\omega|d_n, \tau, T)$ is found using Bayes' rule (3). When the prior distribution $p_{n-1}(\omega)$ is a bimodal Gaussian distribution, with parameters $\mu_{n-1}$ and $\sigma_{n-1}$, analytic expressions can be found for $\langle\omega^2\rangle_n$ and $\langle\omega^4\rangle_n$, due to the simple form of the likelihood function [see Eq. (4)],

$$\langle\omega^2\rangle_n = \frac{\Re\{f_{n-1}(0) + (-1)^{d_n} f_{n-1}(\tau)\}}{\Re\{n_{n-1}(0) + (-1)^{d_n} n_{n-1}(\tau)\}},\tag{8}$$

$$\langle\omega^4\rangle_n = \frac{\Re\{g_{n-1}(0) + (-1)^{d_n} g_{n-1}(\tau)\}}{\Re\{n_{n-1}(0) + (-1)^{d_n} n_{n-1}(\tau)\}},\tag{9}$$

with

$$f_n(t) = e^{-\frac{1}{2}\alpha_n^2 t^2}(\zeta_n^2 + \sigma_n^2)e^{i\mu_n t},\tag{10}$$

$$g_n(t) = e^{-\frac{1}{2}\alpha_n^2 t^2}(\zeta_n^4 + 6\zeta_n^2\sigma_n^2 + 3\sigma_n^2)e^{i\mu_n t},\tag{11}$$

$$n_n(t) = e^{-\frac{1}{2}\alpha_n^2 t^2}e^{i\mu_n t},\tag{12}$$

where $\zeta_n = \mu_n + i\sigma_n^2 t$ and $\alpha_n^2 = \sigma_n^2 + 2T^{-2}$. We then pick the bimodal Gaussian that has its first two non-zero raw moments closest to $\langle\omega^2\rangle_n$ and $\langle\omega^4\rangle_n$, from which the fit parameters $\hat{\sigma}^2$ and $\hat{\mu}$ follow as

$$[\hat{\sigma}_n^{(d_n)}]^2 = \langle\omega^2\rangle_n - \Re\left\{\sqrt{\tfrac{1}{2}\left(3\langle\omega^2\rangle_n^2 - \langle\omega^4\rangle_n\right)}\right\},\tag{13}$$

$$\hat{\mu}_n^{(d_n)} = \sqrt{\langle\omega^2\rangle_n - [\hat{\sigma}_n^{(d_n)}]^2}.\tag{14}$$

In these expressions, we discard imaginary contributions, which can arise when the computed raw moments of the posterior do not adhere to constraints set on the moments of the bimodal Gaussian, occurring when $\mu$ is small compared to $\sigma$. Formally speaking, we see that the calculated value of $\hat{\sigma}_n^2$ becomes complex when the posterior distribution is leptokurtic, which indeed happens when the contributions at positive and negative $\omega$ are not well-separated anymore. Discarding the imaginary part of $\hat{\sigma}_n^2$ in that case corresponds to approximating the posterior distribution by a unimodal Gaussian centered at zero. This approach thus resembles a simplified version of the scheme presented in Ref. [35], where the number of modes in a fitted multi-modal Gaussian needed to be continuously adjusted based on the weight distribution over the modes.

As explained above, the fit parameters $\hat{\sigma}^2$ and $\hat{\mu}$ should in principle be evaluated for each candidate $\tau$, and the optimal waiting time $\tau_n$ will be the one that minimizes the expected variance

$$\mathbb{E}_{d_n}\big[\hat{\sigma}_n^2\big] = \frac{1}{2}\Big\{ \big([\hat{\sigma}_n^{(0)}]^2 - [\hat{\sigma}_n^{(1)}]^2\big)e^{-\frac{1}{2}\alpha_{n-1}^2\tau^2}\cos(\mu_{n-1}\tau) + [\hat{\sigma}_n^{(0)}]^2 + [\hat{\sigma}_n^{(1)}]^2 \Big\}, \tag{15}$$

i.e., $\tau_n = \arg\min_\tau \mathbb{E}_{d_n}[\hat{\sigma}_n^2]$. This quantity is also known as the *risk*, and is found by multiplying the two possible variance outcomes in Eq. 13 by their probability $p(d|\mu_{n-1}, \sigma_{n-1}, \tau, T)$ of occurring based on our current knowledge (found by integrating the prior multiplied with Eq. 4 over all $\omega$'s). The fact that we have, via Eqs. (8–15), an explicit expression for $\mathbb{E}_{d_n}[\hat{\sigma}_n^2]$ allows in principle for minimization of the expected variance. However, since this expression is in general a complicated function of $\tau$ with many local minima, analytic minimization is still challenging and most likely too complex to perform efficiently on an FPGA in real time. Therefore, we start by investigating the limits of small and large $\mu_{n-1}/\sigma_{n-1}$.

For $\mu_{n-1}/\sigma_{n-1} \gg 1$ we find

$$\mathbb{E}_{d_n}\big[\hat{\sigma}_n^2\big] = \sigma_{n-1}^2 - \frac{\sigma_{n-1}^4\tau^2\sin(\mu_{n-1}\tau)}{e^{\alpha_{n-1}^2\tau^2} - \cos(\mu_{n-1}\tau)}. \tag{16}$$

This expression displays fast oscillations as a function of $\tau$, its local minima occurring at times for which $\mu_{n-1}\tau = (k + \frac{1}{2})\pi$, with $k$ an integer. The oscillations have an envelope function $\sigma_{n-1}^2(1 - \sigma_{n-1}^2\tau^2 e^{-\alpha_{n-1}^2\tau^2})$ that is minimal for $\tau = 1/\alpha_{n-1}$. In the limit $\mu_{n-1}/\sigma_{n-1} \gg 1$, the optimal waiting time $\tau_n$ can thus be taken to be

$$\tau_n = \left( \left\lfloor \frac{\mu_{n-1}}{\pi\alpha_{n-1}} - \frac{1}{2} \right\rceil + \frac{1}{2} \right) \frac{\pi}{\mu_{n-1}}, \tag{17}$$

where $\lfloor\ldots\rceil$ denotes rounding off to the nearest integer. This choice of waiting time leads to an expected variance $\mathbb{E}_{d_n}[\hat{\sigma}_n^2] \approx \sigma_{n-1}^2[1 - (\sigma_{n-1}^2/e\alpha_{n-1}^2)]$, cf. Ref. [34].

For the case of $\mu_{n-1}/\sigma_{n-1} \ll 1$ we find

$$\mathbb{E}_{d_n}\big[\hat{\sigma}_n^2\big] = \sigma_{n-1}^2 - \sigma_{n-1}^4\tau^2\Re\left\{ \sum_{\eta=\pm 1} \frac{\sqrt{2 + \eta e^{\frac{1}{2}\alpha_{n-1}^2\tau^2}}}{2\sqrt{2}e^{\frac{1}{2}\alpha_{n-1}^2\tau^2}} \right\}, \tag{18}$$

which has its global minimum at $\tau \approx 1.75/\alpha_{n-1}$, where $\mathbb{E}_{d_n}[\hat{\sigma}_n^2] \approx \sigma_{n-1}^2[1 - 0.60(\sigma_{n-1}^2/\alpha_{n-1}^2)]$. However, we see that if we instead would evaluate the expected variance at $\tau = 1/\alpha_{n-1}$, i.e., at the optimal time we found in the large-$\mu_{n-1}$ limit, we would find an expected variance of $\mathbb{E}_{d_n}[\hat{\sigma}_n^2] \approx \sigma_{n-1}^2[1 - 0.54(\sigma_{n-1}^2/\alpha_{n-1}^2)]$, the improvement in $\hat{\sigma}^2$ being reduced by only 10%. We take this as a motivation to consistently aim for $\tau_n = 1/\alpha_{n-1}$ for the next experiment, throughout the whole range of $\mu_{n-1}/\sigma_{n-1}$.

We will thus always use Eq. (17) for evaluating the new waiting time $\tau_n$, thus picking the local minimum closest to $\tau = 1/\alpha_{n-1}$. However, when $\mu_{n-1} < \frac{1}{2}\pi\sigma_{n-1}$ (which signals

that the oscillations as a function of $\mu_{n-1}\tau$ are slower than the $\tau$-dependence of the envelope function and thus start to become irrelevant, the expected variance ultimately converging to the $\mu_{n-1}$-independent expression that Eq. (18) gives) we take $\tau_n = 1/\alpha_{n-1}$.

Since the method presented in this section contains several approximations and is based on a relatively rough fitting technique, it will not always yield truly optimal fit parameters nor the most effective $\tau_n$. However, as argued above, we expect the results to always be reasonably good and the advantage of the method is that all calculations that need to be done after each FID experiment [i.e., evaluating Eqs. (13,14,17)] amount to evaluating straightforward analytic expressions, which can be done with minimal computational overhead.

## 3.2 KL-divergence fit using a neural network

The problem with MM estimators, while much simpler to calculate than maximum likelihood estimators, is that they in general are biased and not robust with respect to the samples they are derived from. An example is the problem we encounter when getting complex-valued estimators for small $\mu/\sigma$ in the procedure outlined in the previous Section, which is rooted in the fact that we try to estimate parameters of a bimodal Gaussian using samples from a distribution that has higher kurtosis than if they actually were drawn from a bimodal Gaussian. This suggests that in the case of small $\mu/\sigma$, more sophisticated estimators for $\mu$ and $\sigma$ should ideally be used (such as maximum likelihood estimators), typically requiring a numerical fitting procedure. In the context of our work, however, this might be too complex and time-consuming for an efficient real-time implementation.

A possible workaround to investigate is training a neural network to perform the fitting task [54]; indeed, modern FPGAs allow for the implementation of neural networks for on-the-fly processing of data. Ultimately, the problem boils down to mapping the old parameters $\mu_{n-1}$ and $\sigma_{n-1}$ to the optimal waiting time $\tau_n$ and the resulting updated values of $\mu_n$ and $\sigma_n$ for both outcomes $d_n = 0, 1$, i.e., we want to learn the map

$$f : \{\mu_{n-1}, \sigma_{n-1}\} \rightarrow \left\{ \hat{\mu}_n^{(0)}, \hat{\sigma}_n^{(0)}, \hat{\mu}_n^{(1)}, \hat{\sigma}_n^{(1)}, \hat{\tau}_n \right\}. \tag{19}$$

One could thus perform all (computationally costly) numerical fitting beforehand, for a relevant range of parameters $\mu_{n-1}$ and $\sigma_{n-1}$, and then interpolate the map $f$ by teaching it to a neural network.

Here, we investigate this possibility by performing the numerical fit through minimizing the KL-divergence between the true posterior $p_n(\omega|d_n, \tau_n, T)$ and the bimodal Gaussian distribution (6). Explicitly, this is done by finding

$$\mu_n^{(d_n)}, \sigma_n^{(d_n)} = \arg \min_{\mu, \sigma} \int d\omega \, p_n(\omega|d_n, \tau_n, T) \log \left[ \frac{p_n(\omega|d_n, \tau_n, T)}{q(\omega, \mu_{n-1}, \sigma_{n-1})} \right]. \tag{20}$$

The reason we use the KL-divergence, rather than a least-squares fit, is twofold: (i) the KL-divergence is exactly meant as a metric for similarity between two distributions, and (ii) the least-squares fit empirically results in too narrow distributions that have a near-zero probability density in areas where the true posterior actually has a significant weight. In the KL-divergence fit, the second issue is counteracted by the argument of the logarithm, forcing the fitted distribution to cover the true posterior to a greater extent. We emphasize, to avoid confusion, that we are not using the KL-divergence as a loss function for the training of the neural network, but rather to calculate the map (19) to be taught.

The data set for training the neural network is generated on a grid of linearly spaced $\mu_{n-1}$ and log-spaced $\sigma_{n-1}$. We numerically calculate the full posterior distribution for each pair of parameters, for different measurement outcomes $d$ and times $\tau_n$. A numerical KL-divergence fit to a bimodal Gaussian as explained above is performed for each combination of inputs, and

the target value for each feature $\{\mu_{n-1}, \sigma_{n-1}\}$ is chosen to be the set $\{\mu_n^{(0)}, \sigma_n^{(0)}, \mu_n^{(1)}, \sigma_n^{(1)}, \tau_n\}$ that minimizes the expected variance $\mathbb{E}_{d_n}[\sigma_n^2]$ as a function of $\tau_n$.

For this task, we used a standard feed-forward NN while keeping the storage size of a typical FPGA as a boundary condition in mind. The network is trained by minimizing the mean square error (MSE) between predictions and the target values. A subtlety to address is that because of their role in the estimation scheme, the tolerance for errors in the output of the NN varies across the map. Indeed: (i) the errors in $\mu_n^{(d)}$ and $\sigma_n^{(d)}$ must be contained so that the bimodal Gaussian based on the output parameters still has a significant overlap with the one based on the target values, i.e., both errors should not exceed the scale of $\sigma_n^{(d)}$ itself, and (ii) the error in $\tau_n$ must be contained so that the measurement performed using this time is consistent with the updates for $\mu_n^{(d)}$ and $\sigma_n^{(d)}$. In practice, the calculation of a MSE-based loss function therefore requires including a variable weight for the error, depending on the values of the inputs $\mu_{n-1}$ and $\sigma_{n-1}$. We implemented this by instead teaching the NN a map where all five output parameters are renormalized by $\sigma_{n-1}$, using a plain MSE as the loss function, and finally retrieving the predictors of interest by applying the inverse transformation to the output as a post-processing step, see Appendix A for more details.

We found that a sufficiently large neural network is capable of learning an accurate fit over several orders of magnitude of $\mu$, $\sigma$ and $\tau$, though its performance in different regions of the map was not consistent. However, for a truly useful fit, the size of the network needed makes it unfeasible to implement straightforwardly on the current generation of FPGAs. Therefore we explored alternative approaches as well, where the NN is only used in the regime of small $\mu/\sigma$, where the simple MM fit of Section 3.1 does not work optimally. We found that a significantly smaller neural network (three layers of 20 neurons each) manages to consistently learn the map where $\mu < 2\pi\sigma$, and we therefore investigated a hybrid approach as alternative to the MM: we use the MM for $\mu_{n-1} \geq 2\pi\sigma_{n-1}$, while using the NN map when $\mu_{n-1} < 2\pi\sigma_{n-1}$ where a good fit is achievable (cf. Ref. [34]). Although we believe that the network can be made even more compact and still give acceptable results, we did not investigate this further.

## 4 Results

In this Section we analyze the performance of the two estimation schemes outlined above, and compare them to the more commonly used Bayesian protocol with linearly spaced $\tau_n$ [17, 18, 25]. To do so, we first discuss the experimentally relevant time scales involved in the estimation. Firstly, the Hamiltonian parameter $\omega$ to be estimated can be assumed constant during the estimation protocol only if the typical time scale associated with its variation, i.e., the correlation time $T_c \sim \Gamma^{-1}$, is much longer than the total estimation time $T_e \ll T_c$. The estimation time $T_e$ includes $N$ repetitions of the FID experiment, each of which involves an initialization–evolution–readout sequence. During the $n$-th repetition, the qubit undergoes free evolution for time $\tau_n$, while the initialization and readout steps take additional time $T_{\exp}$, typically of the order $T_{\exp} \sim 10\,\mu s$. In total, the $n$-th repetition thus takes $T_n = \tau_n + T_{\exp}$, and the estimation time can be formally written as $T_e = NT_{\exp} + \sum_n \tau_n$. The constraint $T_e \ll T_c$ thus implicitly sets a limit to the number of available repetitions $N$ (in Section 4.2 we will investigate this constraint in more detail). Furthermore, the phenomenological dephasing time $T$ sets the upper bound on the individual evolution times $\tau_n \lesssim T$ and thus limits the total estimation time $T_e \lesssim N(T + T_{\exp})$ as a result.

In our simulations, we first set $T_c \to \infty$, i.e., we treat $\omega$ as a static parameter, and we compare the estimation methods both in the dephasing-free case and for finite $T$. Next, we include the dynamics of $\omega$ by using a finite $T_c$ and we analyze their effect on the estimation procedure, focusing on the relevant example of low-frequency fluctuations of $\omega$ (assuming it

to be the Overhauser field gradient). Finally, we extend this analysis to include an arbitrary additional separation time $T_w$ that elapses between consecutive runs of the estimation protocol and thus defines a time window in which the knowledge obtained about $\omega$ can be employed for qubit control with improved coherence. Altogether, this thus presents a complete protocol for the tracking of a slowly varying Hamiltonian parameter in practice, with small enough overhead to be implemented on an FPGA in a typical experiment.

## 4.1 Estimation of a static parameter

In order to benchmark the schemes, we simulate many estimations where the true parameter $\omega$ to be estimated is assumed static and is drawn from the normal distribution $p_0(\omega)$, truncated here to $\omega \in [-2\sigma_K, 2\sigma_K]$ since the NN is only trained to be valid on this domain, and we use both estimation methods outlined in Section 3. The results following from the MM (red) and hybrid schemes (green) are shown in Fig 3, where we compare them to a standard non-adaptive scheme with uniform sampling in time, $\tau_n = n\pi\sigma_K^{-1}/2$, and a uniform initial prior (blue), as used in [17, 18, 25]. This is the lowest linear sampling rate (longest time spacing) that ensures no aliasing with the prior width $\sigma_K$, although otherwise no optimization of this choice has been made. Other non-adaptive sampling strategies could have been chosen; for instance, it has been long known that exponential sampling can perform better [55]. We have however limited our comparison to linear sampling due to its prevalence in experiments. The distributions of deviations of the estimates $\hat{\mu}_n$ from the true values are plotted as "violin plots" on a logarithmic scale, where the horizontal bars indicate the median of each distribution.[1] Fig 3(a) shows the results as a function of the total number $N$ of FID experiments per estimate, and Fig 3(b) presents the same results but now as a function of the total estimation time $T_e$, where we included $T_{\text{exp}} = 750\,\sigma_K^{-1}$. The inset in Fig 3(a) shows the total number of parameters that need to be stored on the FPGA during an estimation procedure, as a function of the number of experiments each estimate consists of.

As can be seen from Fig 3(a), the median error of both the MM (red) and the hybrid scheme (green) decreases exponentially with the number of measurements $N$ starting at around 20 measurements, and both schemes clearly outperform the uniform sampling (blue). This exponential improvement is similar to the one found in Ref. [34], where a two-step process was employed, consisting of a "warm-up" round of estimates with linearly spaced $\tau_n$, followed by an adaptive procedure similar to the one outlined in Section 3.1 where $\mu/\sigma$ was assumed to have become large enough such that the positive part of the distribution could be fitted straightforwardly to a single Gaussian peak. Having such a warm-up round means that the scheme does not take advantage of the fact that the Gaussian approximation only requires storing two variables since the warm-up needs to store the entire distribution. Also, for very small $\omega$ one would presumably have to use many measurements in the warm-up period.[2]

Since the two adaptive schemes are optimized in a greedy way, i.e., to yield the maximum gain per experiment, they show their exponential improvement clearly as a function of $N$. The improvement is less pronounced when considering the total experiment time $T_e$ rather than the number of experiments, as illustrated in Fig 3(b). Indeed, greedy schemes typically tend to require exponentially spaced waiting times, and 50 experiments with such an adaptive scheme will thus take significantly longer than the same number of linearly spaced experiments. However, since the experimental overhead time $T_{\text{exp}} \sim 10\,\mu s$ needed for initialization and readout

---

[1]The reason for plotting the median rather than the mean is that the latter will be skewed to deceivingly large values due to the occurrence of few outliers where the estimation fails [56].

[2]Asymptotically, all of the adaptive optimization schemes mentioned implement a quantum binary search algorithm where the binary choices relate to the measurement outcomes $d = 0$ and $d = 1$. Each outcome results in a different posterior, choosing which side of the prior's mean it centres on, giving a factor $1-e^{-1} \approx 0.63$ improvement in the variance at each step, cf. Eq. (16).

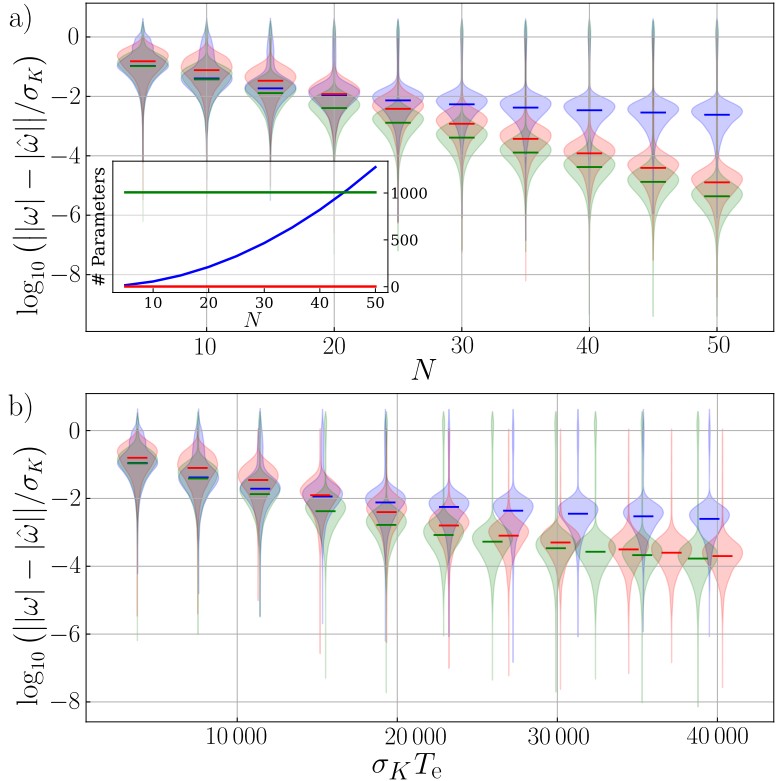

Figure 3: (a) Violin plots of the logarithm of the absolute estimation errors of $10^4$ simulations after $N$ single-shot measurements with linear $\tau_n$ (blue) and adaptive $\tau_n$ using the MM scheme (red) or the hybrid scheme (green), where the true frequency is sampled from the initial prior truncated to $\omega \in [-2\sigma_K, 2\sigma_K]$. Horizontal lines indicate the median error over the simulations. The inset shows for each of the schemes how many parameters have to be stored during the estimation. (b) Same results versus total experiment time $T_e$ assuming that initialization and readout take $T_{\exp} = 750\,\sigma_K^{-1}$.

is typically orders of magnitude longer than $\sigma_K^{-1} \sim 20$ ns, which sets a typical scale for the waiting times, the number of experiments $N$ and the total time $T_e$ often scale similarly for moderate $N$, typically up to tens of experiments, see Fig 3. Interestingly, the scheme based on the MM does not fare very well for small numbers of measurements (up to $N \sim 15$), likely due to the approximation for evaluating the evolution time being suboptimal at small $\mu/\sigma$. We also note that, although the NN interpolation of a KL-divergence fit gives an optimal choice for the waiting times also at small $\mu/\sigma$, it does not seem to give a significant improvement over the uniform time spacing during the first few measurements.

The inset of Fig 3(a) gives an indication of the amount of memory needed to perform the different schemes, as a function of $N$. The scheme based on the MM (red) only requires tracking of *three* parameters (and calculating very few equations). While the hybrid scheme (green) technically only needs to track seven variables, it does need to store the NN on the FPGA and to feed data through the network (which in this case consisted of 1005 trainable parameters). For the uniform time sampling (blue) the computational cost depends on how the procedure is implemented. Here, we used a Fourier-coefficient representation of the instantaneous distribution functions [33], so that the number of parameters increases quadratically with $N$ and the distribution is represented accurately at all times.[3] Comparing the three meth-

---

[3]One could alternatively represent the distribution by discretizing $\omega$ with a desired resolution or use Monte-

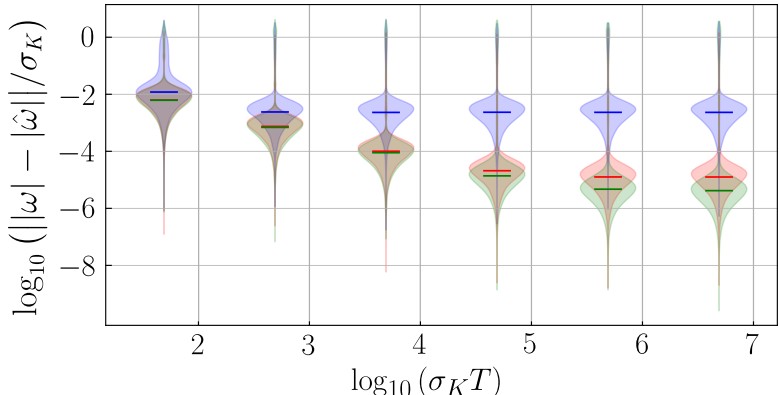

Figure 4: Violin plots of the logarithm of the absolute estimation errors of $10^4$ simulations after $N = 50$ single-shot measurements as a function of the phenomenological dephasing time $T$. We show results with linearly spaced waiting times $\tau_n$ (blue) and adaptive $\tau_n$ using the MM scheme (red) or the hybrid scheme (green), where the true $\omega$ is sampled from the initial prior truncated to $\omega \in [-2\sigma_K, 2\sigma_K]$. Horizontal lines indicate the median error over the simulations.

ods, the computational advantage of the simple MM approach is dramatically clear and, in our opinion, clearly outweighs the modest loss in accuracy for smaller $N$.

We now add a finite dephasing time $T$, which limits the evolution time during each repetition, $\tau_n \lesssim T$. In its simplest form, the phenomenological parameter $T$ corresponds to the time scale on which the coherent oscillations that are probed in each FID experiment preserve. As an example, for the case where coherence is limited by qubit relaxation or by leakage to non-computational states, the parameter would be directly related to the relaxation or leakage time $T \sim T_1$. Alternatively, the time scale $T$ can be related to the fluctuations of the parameter to be estimated itself around its static value; if such fluctuations would consist of uncorrelated white noise, then the dephasing time is simply $T \sim T_2$, where $T_2$ is the commonly measured dephasing time.[4]

To show how the value of $T$ affects the estimation procedure, we plot in Fig 4 the distributions of absolute errors after $N = 50$ FID experiments as a function of the normalized dephasing time $T\sigma_K$ for the same three schemes as in Fig 3, using the same color coding. The distributions are again plotted as violin plots on a logarithmic scale, where the horizontal lines indicate the median. As the coherence time $T$ becomes short, the advantage of the adaptive schemes diminishes and the accuracy eventually becomes similar to that of the uniform time sampling. Exponential improvement of the error is only possible with an exponential increase in waiting times $\tau_n$, and since the FID times are capped at $\tau_n \lesssim T$ the usefulness of adaptive schemes naturally becomes limited in the case of very short dephasing times.

## 4.2 Single estimation of a slowly drifting parameter

For the case of $\omega$ representing the slowly drifting Overhauser field gradient, with a typical correlation time $T_c \sim 5$ s, the longest useful waiting time (related to the dephasing time $T$ discussed above) becomes more intricately connected to the other time scales mentioned

---

Carlo sampling to get an arbitrarily good resolution without storing too many data points.

[4]We highlight that in all of the above cases the decay of the likelihood function [cf. Eq. (4)] would be exponential in $T$, instead of Gaussian. This would, however, only modify details in the estimators $\hat{\sigma}_n^{(d_n)}$ and $\hat{\mu}_n^{(d_n)}$ but keep the intuitive interpretation of $T$ as the upper bound of $\tau$ intact. In this work we chose Gaussian decay, to align with the case of spin qubits affected by low-frequency noise, in the form of the fluctuating Overhauser fields themselves or possibly as residual effects on $\omega$ of $1/f$ charge noise.

in the beginning of this Section. Assuming that the dynamics of $\omega$ are fully driven by the fluctuations of the nuclear spin ensemble, which we model as an OU process, we investigate the effect of these drift–diffusion dynamics on the estimation procedure.

In the absence of any additional information about the Overhauser gradient $\omega$, our knowledge about it is described by the probability distribution $p_0(\omega)$. The uncertainty associated with this distribution then determines the "standard" dephasing time usually associated with the fluctuating Overhauser fields in GaAs-based spin qubits, $T_2^K \sim \sigma_K^{-1} \sim 20$ ns. The role of our estimation procedure is to reduce the uncertainty in $\omega$ to a final value $\sigma_f \ll \sigma_K$, and hence significantly extend the dephasing time of the qubit for operations performed right after the estimation. One can clearly not reach a final uncertainty $\sigma_f$ if $\omega$ diffuses over more than $\sim \sigma_f$ during a single FID experiment.[5] To estimate the time $\tau_{2\sigma_f}$ over which a Gaussian peak with width $\sigma_f$ evolves to a peak with width $2\sigma_f$ we can use Eq. (2), yielding[6]

$$\tau_{2\sigma_f} \sim \frac{T_c}{2} \ln\left( \frac{\sigma_K^2 - \sigma_f^2}{\sigma_K^2 - 4\sigma_f^2} \right). \tag{21}$$

In the limit of $\sigma_f \ll \sigma_K$, this reduces to

$$\tau_{2\sigma_f} \sim \frac{3T_c}{2} \frac{\sigma_f^2}{\sigma_K^2}. \tag{22}$$

Since both adaptive estimation schemes investigated in this work are greedy and converge to a roughly exponential increase of $\tau_n$ and exponential decrease of $\sigma_n$ as a function of the experiment number $n$, the relation (22) also yields a maximal achievable accuracy $\sigma_{\min}/\sigma_K$ and corresponding maximum number of useful single-shot experiments $N_{\max}$ as a function of $T_c$. In principle, these quantities are defined through $\tau_{N_{\max}} + T_{\exp} = \tau_{2\sigma_{\min}}$, but throughout this section we will set $T_{\exp} = 0$, in order to find the intrinsic limits on the estimation accuracy purely set by the dynamics of $\omega$ interfering with the FID experiments.

The expectation is that as soon as the two Gaussian peaks in the bimodal distribution are well separated, the schemes will converge to a sequence where $\tau_n \sim a_\tau (1 - e^{-1})^{-n/2}$ and $\sigma_n \sim a_\sigma (1 - e^{-1})^{n/2}$ [35], where the prefactors $a_\tau$ and $a_\sigma$ depend on how quickly this exponential regime is reached. Using Eq. (22) we find

$$N_{\max} \approx \frac{2 \ln\left( \frac{3}{2} \frac{T_c}{a_\tau} \frac{a_\sigma^2}{\sigma_K^2} \right)}{3[1 - \ln(e - 1)]}. \tag{23}$$

This sets the minimal achievable variance as $\sigma_{\min}^2 \sim a_\sigma^2 (1 - e^{-1})^{N_{\max}}$, for which the longest single-shot waiting time needed is $\tau_{N_{\max}} = a_\tau (1 - e^{-1})^{-N_{\max}/2}$. Since this time $\tau_{N_{\max}}$ is in any case the longest useful waiting time, one can use $T = \tau_{N_{\max}}$ to limit the choice of $\tau_n$ accordingly. The presence of a finite $T_{\exp}$ can be incorporated straightforwardly into the theory, resulting in results that are slightly modified quantitatively.

To illustrate this intrinsic limitation on the estimation accuracy, we simulated $10^4$ estimations up to $N = 60$ using the scheme based on the MM, while letting $\omega$ continuously fluctuate

---

[5]If this constraint is violated, then the estimation procedure might produce so-called outliers, i.e., estimates that are much further off from the true value of $\omega$ than typical ones.

[6]Formally we always work with the bimodal distribution $q(\omega, \mu, \sigma)$, but since both this distribution and the dynamics of $\mu$ and $\sigma$ following from the FP equation are symmetric in $\omega$, we can simply use Eq. (2) to predict the evolution of our parameters. Furthermore, the relaxation of $\mu(t)$ toward zero also contributes to the drift of the distribution function describing $\omega$. For small $\sigma_f/\sigma_K$ and $t/T_c$, however, we find that this relaxation goes $\propto t/T_c$, whereas the change in $\sigma$ is $\propto (t/T_c)^{1/2}$.

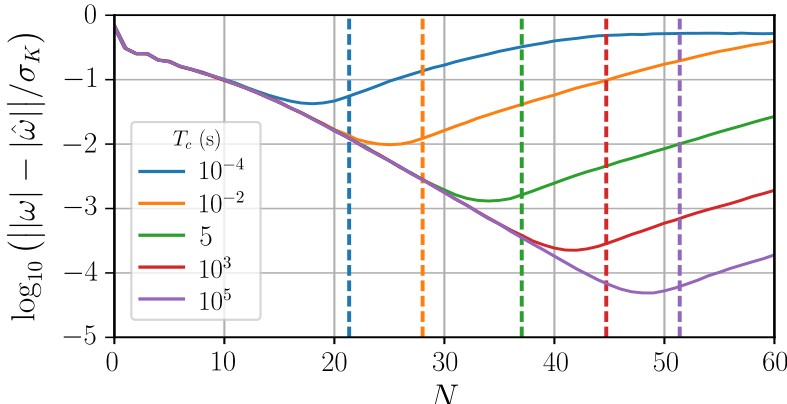

Figure 5: Median estimation error as a function of the number of single-shot experiments used, for $10^4$ simulations of the scheme based on the MM. Here, we included the drift–diffusion dynamics of $\omega$ throughout the whole simulation, using $\sigma_K = 50$ MHz and different correlation times $T_c$ (different traces). The vertical dashed lines show the maximum useful number of single shots as given by Eq. (23).

following an OU process with $\sigma_K = 50$ MHz and $T_{\exp} = 0$. In Fig 5 we plot the median accuracy of the resulting estimates as a function of the number of total FID experiments,[7] for different $T_c$ ranging from 100 $\mu s$ to $10^5$ s (different traces). We found that for our simulations, Eq. (23) predicts that $N_{\max} \approx 1.45 \ln(494 \, \sigma_K T_c)$, where the factors $a_\tau$ and $a_\sigma$ were determined from fitting $a_\tau (1 - e^{-1})^{-n/2}$ and $a_\sigma (1 - e^{-1})^{n/2}$ to the average of actual $\tau_n$ and $\sigma_n$ used by the scheme, in the limit of $T_c \to \infty$. The resulting values for $N_{\max}$ are indicated in Fig 5 with vertical dashed lines, confirming that Eq. (23) gives a good estimate for the maximum useful number of FID experiments. This means that for a typical spin-qubit system (where $T_c \approx 5$ s) and using the greedy adaptive estimation schemes presented in this paper, there is generally no point in performing more than $N \sim 35$ single-shot experiments in a single estimation, and the dephasing time can be set to $T \sim \tau_{35} \approx 3 \, \mu s$. This also means that the intrinsic limitation on the estimation accuracy caused by the diffusion of $\omega$ is roughly $\sigma_{\min}/\sigma_K \approx 10^{-3}$.

## 4.3 Sequential estimation of a slowly drifting parameter

Another consequence of having an $\omega$ that slowly drifts is that in practice the estimation procedure has to be repeated over time to update our knowledge about $\omega$, in order to keep the effective qubit dephasing time suppressed. Denoting the time in between two estimations by $T_w$, we understand that if $T_w \gtrsim T_c$ all previously gained knowledge about $\omega$ has become obsolete at the start of each estimation procedure and the correct initial prior is always $p_0(\omega)$. However, if one sets $T_w \lesssim T_c$ then we can expect the initial prior to be still somewhat narrowed, potentially allowing for a more efficient accurate estimation of $\omega$ using fewer FID experiments.

An important advantage of using the Gaussian approximation throughout the whole estimation scheme is that it is very straightforward to include such an "idle time" into the model. Indeed, if the final posterior distribution of the estimation can be mapped to the Gaussian parameters $\mu_f$ and $\sigma_f$, then we can use Eq. (2) to obtain the parameters that characterize the distribution at time $T_w$ after the estimation procedure,

$$\mu(T_w) = \mu_f e^{-T_w/T_c}, \tag{24}$$

$$\sigma(T_w) = \sqrt{\sigma_K^2 + (\sigma_f^2 - \sigma_K^2)e^{-2T_w/T_c}}. \tag{25}$$

---

[7]Defining the "correct" value to be $\omega(t)$ at the end of the final FID experiment of each estimation.

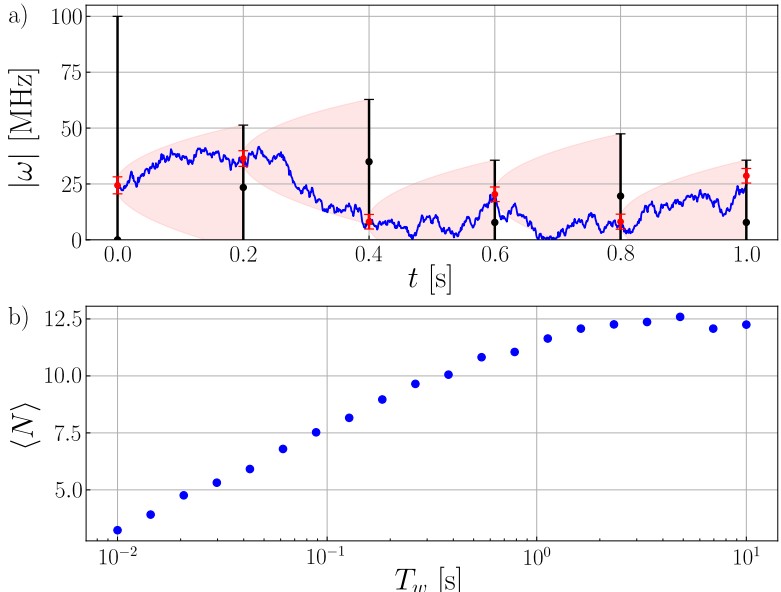

Figure 6: (a) Example of tracking a drifting Overhauser gradient (shown in blue) with the hybrid estimation scheme. A series of FID experiments is performed every 200 ms until the final Gaussian posterior variance becomes $\sigma_f < 2$ MHz (assuming that $\omega$ remains constant within these few measurements). Red dots mark the final posterior estimates $\mu_f$, with error bars of $2\sigma_f$. Equations (24,25) give a recipe for how to adjust the new prior as the time between estimations increases (the shaded red regions indicate the evolution of $\mu$ and $2\sigma$). The black dots show the resulting prior estimates $\mu(T_w)$, again with error bars of $2\sigma(T_w)$. (b) The average number of single shots needed to regain a final posterior variance with $\sigma_f < 2$ MHz as a function of the idle time $T_w$ between measurements.

We see that when $T_w \lesssim T_c$ the final $\sigma(T_w)$ is indeed significantly smaller than $\sigma_K$. The values for $\mu(T_w)$ and $\sigma(T_w)$ as given by Eqs. (24,25) can easily be evaluated on an FPGA, allowing to start the subsequent estimation from the initial prior $q[\omega, \mu(T_w), \sigma(T_w)]$ instead of $p_0(\omega)$. In a way, we thus keep track of the optimal prior to use, which presumably reduces the number of FID experiments needed in each estimation to achieve good accuracy.

In Fig 6 we illustrate this approach, using the hybrid estimation scheme presented in Section 3.2 and setting $T_c = 5$ s and $\sigma_K = 50$ MHz. Fig 6(a) shows a simulated $\omega(t)$ following from an OU stochastic process in blue. We then simulated six subsequent estimation procedures, spaced by $T_w = 0.2$ s, where we stopped whenever we reached $\sigma_f \leq 2$ MHz. The resulting final values $\mu_f$ and $\sigma_f$ are indicated by the red points with error bars (the error bars show $2\sigma_f$). The initial prior at $t = 0$ is given by $p_0(\omega)$, depicted as the black point at $|\omega| = 0$ with an error bar of $2\sigma_K$. The evolution of $\mu$ and $2\sigma$ as given by Eqs. (24,25) in between estimations is illustrated by the red shaded areas, still resulting in initial priors (shown in black) for all estimations after the first one that are significantly narrower than $p_0(\omega)$. We find that for the values used here, all subsequent estimations require $N \approx 9$ to obtain an accuracy of $\sigma_f = 2$ MHz, whereas one typically needs $N \approx 13$ to reach the same accuracy starting from $p_0(\omega)$. In Fig 6(b) we explore the dependence of the average number of FID experiments needed, $\langle N \rangle$, to reach $\sigma = 2$ MHz on the idle time $T_w$ in between estimations. We see that $\langle N \rangle$ increases roughly logarithmically until $T_w \sim T_c$, where it saturates at $\langle N \rangle \approx 12.5$.

Depending on the time window needed for coherent qubit operations, one could thus efficiently reduce the estimation overhead needed by adjusting $T_w$ in the experiment and using

Eqs. (24,25) for an adaptive adjustment of the initial prior for each estimate. Conversely, one can use the results presented in Section 4.2 to estimate the maximal time $T_w$ in between estimations for which the uncertainty in $\omega$ stays below a given threshold. Suppose that all qubit operations require $\sigma < \sigma_{\max}$ and that one is able to efficiently estimate $\omega$ to an accuracy $\sigma_f$; then the time window available for coherent qubit operations is given by

$$T_w \sim \frac{T_c}{2} \ln\left( \frac{\sigma_K^2 - \sigma_f^2}{\sigma_K^2 - \sigma_{\max}^2} \right), \tag{26}$$

which for $\sigma_{\max}, \sigma_f \ll \sigma_K$ reduces to $T_w \sim T_c(\sigma_{\max}^2 - \sigma_f^2)/2\sigma_K^2$.

## 5 Summary and conclusions

We have investigated two different efficient adaptive Bayesian estimation schemes that can track a slowly fluctuating Overhauser field gradient with a zero-mean Gaussian distribution in time, using a series of free induction decay experiments. Both schemes perform a greedy optimization of the single-shot estimation parameters based on the current knowledge of the field gradient, in order to obtain an exponential scaling of the estimation error with the number of experiments. The small number of variables needed by these schemes to track the gradient makes them well-suited for real-time estimation performed on an FPGA, and the robustness of Bayesian methods combined with the ability of the schemes to handle zero-value estimates makes them applicable to real-world quantum estimation problems. We also show how our simple Gaussian approach lends itself excellently for *predictive* estimation by use of the Fokker-Planck equation to anticipate how our knowledge of the field gradient evolves after an estimation has been performed. We included a discussion of the effects of a finite dephasing time on the estimation schemes and we analyzed the effect of the fluctuations of the field gradient itself on the robustness of the schemes, yielding a useful insight in the intricate interplay of all experimental time scales involved.

## Acknowledgments

We gratefully acknowledge useful discussions with Fabrizio Berritta, Torbjørn Rasmussen, Anasua Chatterjee, and Ferdinand Kuemmeth.

**Funding information** This project was funded within the QuantERA II Programme that has received funding from the European Union's Horizon 2020 research and innovation programme under Grant Agreement No 101017733. This work is part of INTFELLES-Project No. 333990, which is funded by the Research Council of Norway (RCN), and it received funding from the Dutch National Growth Fund (NGF) as part of the Quantum Delta NL programme. Parts of the computations were performed on resources provided by the NTNU IDUN/EPIC computing cluster [57].

**Code availability** Examples of code used for the numerical calculation can be found at https://github.com/jacobdben/efficient-bayesian-estimation.

# A Training the neural network

As mentioned in the main text, it becomes necessary to teach the NN a renormalized map to get acceptably bounded errors such that the NN yields outputs that perform the correct update of the information about $\omega$ across a larger range of parameters. Specifically, the renormalized map that we used in teaching the NN reads as

$$\tilde{f} : \{\mu_{n-1}, \sigma_{n-1}\} \rightarrow \left\{ \frac{\hat{\mu}_n^{(0)} - \mu_{n-1}}{\sigma_{n-1}}, \frac{\hat{\sigma}_n^{(0)}}{\sigma_{n-1}}, \frac{\hat{\mu}_n^{(1)} - \mu_{n-1}}{\sigma_{n-1}}, \frac{\hat{\sigma}_n^{(1)}}{\sigma_{n-1}}, \frac{\sigma_{n-1}\hat{\tau}_n}{2} \right\} . \tag{A.1}$$

Data is generated over a grid of inputs $\mu_{n-1}$ and $\sigma_{n-1}$, with a linearly spaced range $[0, 1]$ and logarithmically spaced range $[0.001, 0.5]$ of the values, respectively. For illustration, the desired output of $\tau_n$ and its renormalized counterpart $\sigma_{n-1}\tau_n/2$ are shown in Fig 7(a,b), where we set $T \rightarrow \infty$ for simplicity. It is important to accurately capture the fine features that are visible in the renormalized output [Fig 7(b)] and not in the original output [Fig 7(a)], since one otherwise would perform a measurement that is inconsistent with the updated distribution of $\omega$. In Fig 7(a) these features become obscured by the large range of values the output can have (a difference of two orders of magnitude in this case).

In order to obtain some insight into how the NN is learning the map, we present snapshots of the learning process in Fig 8, using the same feed-forward network with three layers, each with 20 neurons, and hyperbolic tangent activation functions as used in Section 4. The five small plots show a line trace taken at $\mu_{n-1} = 0.5$ of the output $\sigma_{n-1}\tau_n/2$ to be learned (faint red line) together with the learned output (red dashed line) at different stages of the training. The bottom right plot shows the validation loss curve (blue dots) and a quantification of the number of visible peaks in the output of the NN (black line), with this number being an indication of the fineness of the features in the map that the network has learned. The NN first learns the region of small $\mu_{n-1}/\sigma_{n-1}$ in the map [bottom right corner in Fig 7(b)], and then gradually learns features for increasing values of $\mu_{n-1}/\sigma_{n-1}$. This is also reflected in the loss curve, as drops in the loss seem to be correlated with the NN's discovery of a new peak. Of course, the decision for what counts as a visible peak is somewhat subjective; we used the function scipy.signal.find_peaks (with prominence=0.05) to count the number of peaks.

While the NN finds the first few peaks at small $\mu_{n-1}/\sigma_{n-1}$ relatively quickly, the subsequent peaks at larger $\mu_{n-1}/\sigma_{n-1}$ take increasingly longer to learn. It is thus hard to quantify how many peaks are eventually possible for the NN with 3 layers of 20 nodes to learn, in the sense

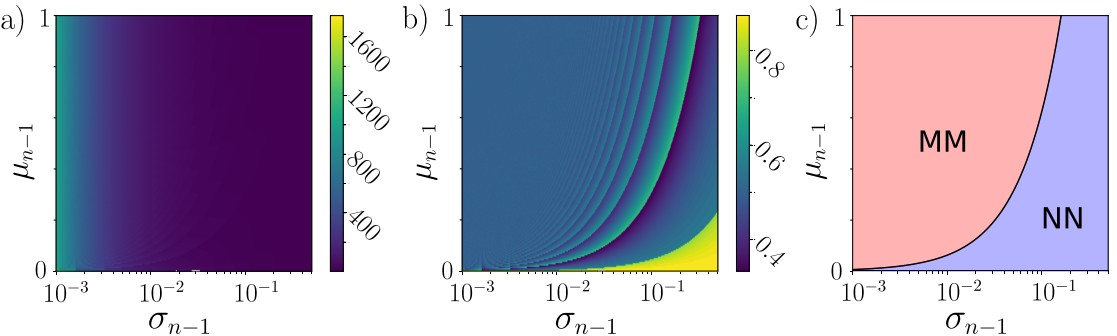

Figure 7: (a) The original target values for $\tau_n$ and (b) the renormalized values $\sigma_{n-1}\tau_n/2$, for $T \rightarrow \infty$. The details that become visible after renormalization are important for consistency between updated parameters. (c) The hybrid estimation scheme partitions the domain of inputs at $\mu_{n-1} = 2\pi\sigma_{n-1}$, where the NN is used for small $\mu_{n-1}/\sigma_{n-1}$ (blue region) and the MM for large $\mu_{n-1}/\sigma_{n-1}$ (red region).

that the number of training epochs needed quickly becomes impractically large. The fact that the region with large $\mu_{n-1}/\sigma_{n-1}$ is difficult to teach the NN was the motivation for the hybrid scheme that only uses the NN fitting where it was able to learn well (small $\mu_{n-1}/\sigma_{n-1}$), while switching to the method of moments fitting where the NN struggles (large $\mu_{n-1}/\sigma_{n-1}$). The decision boundary for the hybrid method was set to $\mu_{n-1} = 2\pi\sigma_{n-1}$, as illustrated in Fig 7(c).

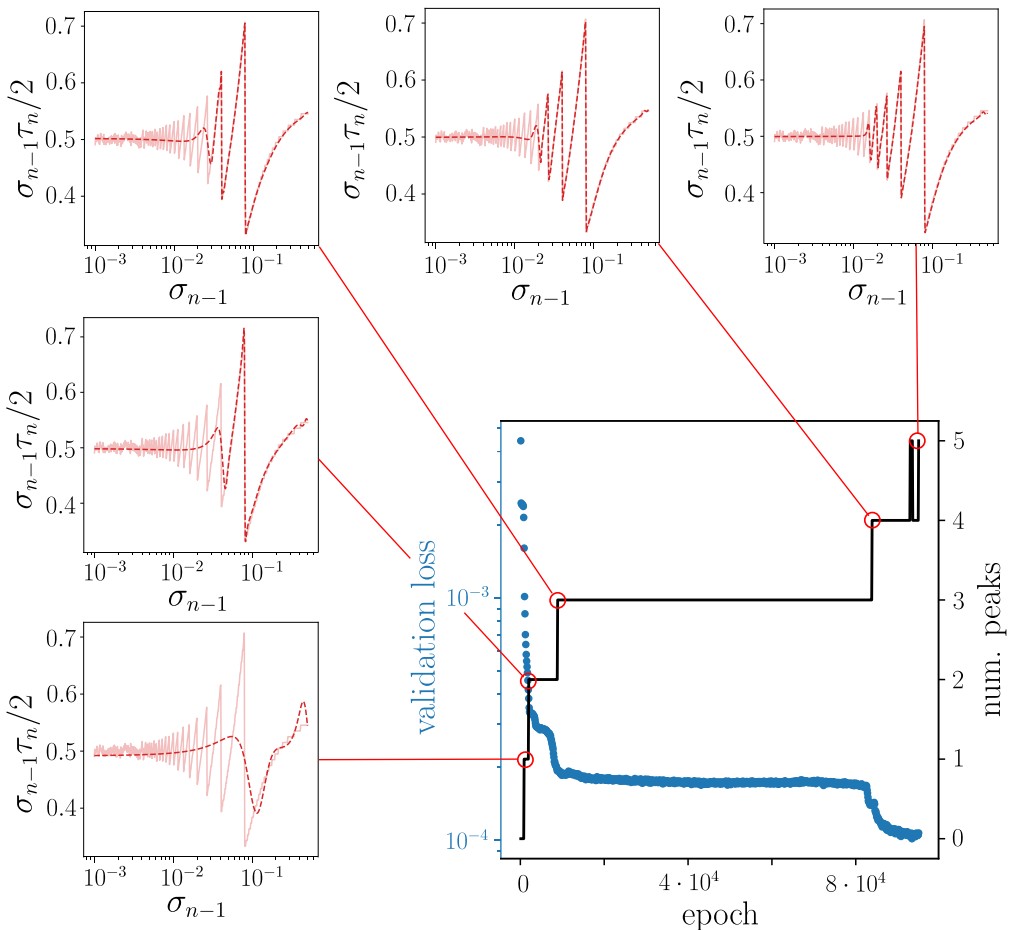

Figure 8: (small plots) Snapshots of the learning process using the renormalized $\tau_n$. The faint read lines show a line cut of the map to be learned at $\mu_{n-1} = 0.5$, the dashed lines the output of the NN at different stages of the training process. (bottom right) Validation loss as a function of training epoch number (blue dots, left) and number of peaks found in the output at $\mu_{n-1} = 0.5$ (black line, right).

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
