# Peer review of "Efficient adaptive Bayesian estimation of a slowly fluctuating Overhauser field gradient"

_SciPost Physics, doi:SciPost Phys. 17, 014 (2024)_

## Round 2 · Referee Report · Anonymous (Referee 1) · 2024-2-12

Report
In singlet-triplet qubits, information about the Overhauser field gradient can be extracted through a free induction decay (FID) experiment, which yields a binary measurement result depending on the field gradient strength and the FID time. The outcome of such a measurement can then be used to perform a Bayesian update on an estimate of the field gradient distribution. The speed at which one can gain certainty about the field gradient crucially depends on the chosen FID times and initial distribution.
In this work, the authors develop a greedy approach that describes the field gradient by a (bimodal) Gaussian distribution and adaptively pick the FID times of the next iteration by minimizing the expected variance. Here the distribution after the application of Bayes’ rule is found through a fit. Using the method of moments for the fit, the authors are able to find approximate analytic expressions for the optimal FID times. Simulating the estimation protocol for a realistic set of parameters, the authors find that it significantly outperforms a standard non-adaptive approach. The authors further improve the performance of the scheme by training a compact neural network to find the optimal FID times in the parameter regime where the approximate analytic solution breaks down. Finally, the authors analyze the optimal number of FID iterations given that the estimated parameter is drifting and describe how previous estimates can be used to improve the initial guess of the gradient field in the next iteration.
Overall, the paper is well written and as far as I can judge scientifically sound. It is also highly relevant as offers an efficient estimation protocol that can be run on a FPGA for fast feedback. I recommend publication and only have some minor comments.
Requested changes
(1) Maybe one could mention that Eq. (15) is just Eq. (4) multiplied with the results in Eq. (13). I was also wondering why T in Eq. (4) is replaced by alpha_n in Eq. (15).
(2) The experiment time Te in Fig. 3 has no unit. Should it be multiplied by sigma_K as in Fig. 4?
We thank the Referee for taking the time to review our submission and we were happy to read their positive recommendation. We address the two comments below:
Comment (1):
Eq. (15) is in fact slightly different than just Eq. (4) multiplied by the results in Eq.~(13).
Since we do not exactly know the value of $\omega$, we must integrate the prior distribution multiplied with Eq. (4) over all $\omega$, in order to get the probability of the two different outcomes, which is then what is multiplied with the results given in Eq. (13).
This subtle difference is also related to the appearing of $\frac{1}{2}\alpha_n = \frac{1}{2}\sigma_n^2 + T^{-2}$ in the exponent in Eq. (15), instead of just the dephasing time $T^{-2}$.
We have now made this more clear by adding the following sentence just after Eq.~(15):
"This quantity is also known as the risk, and is found by multiplying the two possible variance outcomes in Eq. (13) by their respective probabilities $p(d|\mu_{n-1}, \sigma_{n-1}, \tau, T)$ of occurring based on our current knowledge (which is found by integrating the prior multiplied with Eq. (4) over $\omega$)."
Comment (2):
Indeed, the multiplication by $\sigma_K$ in Fig. 3b was missing and has now been added. We thank the Referee for pointing this out.

Author: Jacob Benestad on 2024-06-13 [id 4563]
(in reply to Report 2 on 2024-04-17)We were happy to read the Referee's positive report and thank the Referee for reviewing our manuscript. We address the two comments below:
Comment (1):
That is a good question. Indeed, having phase-control of the Ramsey pulse would give a way to determine the sign of the frequency to be estimated, meaning that one could make the approximation of the prior with a fit to a single Gaussian rather than the bimodal Gaussian which is necessary in our case. This also alleviates the complications when the value to be estimated is very close to zero.
We added the following remark about this in the paragraph below Eq.~(5):
"However, we note that if it is possible to control the cosine phase in Eq. (4) it would be possible to distinguish the sign of $\omega$."
Comment (2):
The Referee is correct that in theory there exist often better choices for non-adaptive sampling strategies than linearly spaced times. We chose the linear strategy as a benchmark because it is currently the most widely used one in experiment and allows for performing a substantial number of FID experiments with relevant waiting times within a realistic dephasing time window $T$. The only way we optimized our non-adaptive method was by making the separation time between measurements as long as possible (since extremely small rotations on the Bloch sphere would provide virtually no information).
We have added the following sentences [with an additional reference to Barna et al., J. Magn. Reson. (1969) 73(1), 6977 (1987)] to the manuscript to make this clear:
"This is the lowest linear sampling rate (longest time spacing) that ensures no aliasing with the prior width $\sigma_K$, although otherwise no optimization of this choice has been made. Other non-adaptive sampling strategies could have been chosen; for instance, it has been long known that exponential sampling can perform better [55]. We have however limited our comparison to linear sampling due to its prevalence in experiments."

---

## Round 2 · Referee Report · Anonymous (Referee 2) · 2024-4-17

Report
One of the issues with real-time sequential Bayesian estimation is the computational burden in determining the best parameters. The two techniques presented in the manuscript reduce the burden by approximating the probability distribution with a bi-modal Gaussian, determined by either computing the lower-order moments of the distribution or interpolating through a neural network.
The paper is clear and well written, the arguments are sound and convincing. I think it clearly deserves publications, and I do not have any major comments.
Just two minor comments/questions:
(1) in Eq 4, would considering the phase of the Ramsey (i.e. by controlling the phase of the second pi/2 pulse) bring any advantages?
(2) the authors compare their adaptive scheme to "standard" (non-adaptive) uniform sampling in time. I think other papers have used linear sampling with different coefficients, or even exponentially increasing times? Can the author discuss whether their choice of non-adaptive sampling is in any way "optimal"?
Recommendation
Publish (easily meets expectations and criteria for this Journal; among top 50%)

---

## Round 3 · Author Response

Thank you for handling our manuscript, and we thank the referees for their helpful comments.

---

## Round 3 · List of Changes

- Fixed the integration limits in Equation 7.
As reply to referee 1: - Added a sentence to elaborate on Eq. (15) - Fixed the label in Figure 3b
As reply to referee 2: - Added a sentence on the possibilities provided by controlling the phase of Eq. (4) - Added three sentences about the choice of non-adaptive sampling. - Added the reference [55]

---

## Editorial Decision

published